# Science vs. Conspiracy Theory about COVID-19: Need for Cognition and Openness to Experience Increased Belief in Conspiracy-Theoretical Postings on Social Media

**DOI:** 10.3390/bs12110435

**Published:** 2022-11-07

**Authors:** Phillip Ozimek, Marie Nettersheim, Elke Rohmann, Hans-Werner Bierhoff

**Affiliations:** 1Department of Psychology, University of Hagen, 58097 Hagen, Germany; 2Department of Psychology, Ruhr University Bochum, 44801 Bochum, Germany

**Keywords:** COVID-19, conspiracy theories, need for cognition, agreeableness, openness to experience, social media

## Abstract

In the context of COVID-19 virus containment, there is a lack of acceptance of preventive measures in the population. The present work investigated which factors influence the belief in scientific propositions compared with belief in conspiracy theories. The focus here was on the determinants of conspiracy beliefs in the context of COVID-19 related media content. Using an online questionnaire (*N* = 175), results indicate that scientific compared to conspiracy-theoretical media content led to higher acceptance. Furthermore, need for cognition (NFC-K), a conspiracy-theoretical worldview (CMQ), and openness to experience (NEO-FFI) were positively associated with conspiracy beliefs derived from Facebook postings. In addition, a conspiracy-theoretical worldview was negatively associated with belief in scientific media content. Furthermore, agreeableness was unrelated to conspiracy beliefs, although it was positively associated with conspiracy-theoretical worldview. The results imply promising persuasion strategies for reducing conspiracy-theoretical beliefs and to increase the acceptance of preventive measures.

## 1. Introduction

As a socially stressful event, the COVID-19 (coronavirus disease 2019) pandemic triggers feelings of powerlessness and being overwhelmed, as well as stress [1,2]. According to Swami and colleagues [3], this leads to a preference for quick and easy explanations for uncertainties. These can be found in conspiracy-theoretical media content. Conspiracy beliefs are defined as false beliefs that find an explanation for an event in a conspiracy with the aim of hiding causes [4]. COVID-19-related conspiracy beliefs are, for example, that COVID-19 presents the vaccination industry as a conspiracy with the profiting involvement of Bill Gates or the virus as a bioweapon that was created on purpose.

On the one hand, conspiracy theories grow more frequently in pandemic contexts [5]. On the other hand, crisis situations such as pandemics increase the recourse to conspiracy-theoretical explanations [6]. This development constitutes, according to the World Economic Forum 2013, a major threat to society [7].

Conspiracy beliefs with respect to the origin and maintenance of COVID-19 impair scientifically based social action more [8]. This is reflected, for example, in reduced vaccination acceptance or reduced adoption of evidence-based prevention and treatment approaches in general [9,10] as well as in relation to COVID-19 [1,11,12,13,14]. To counteract this problem, the development of an empirically confirmed framework of the determinants of belief in conspiracy theories is desirable. To facilitate the interpretation of the results the special emphasis in this study is on the examination of determinants of the belief in conspiracy theories contrasted with the determinants of belief in scientifically grounded media content. 

In the digital age, social media and its reach are of great importance. Increased exposure to COVID-19-related stress-inducing media content presumably creates a downward spiral in the emergence of conspiracy-theoretical beliefs (cf. [11,15]). Social media is furthermore known for using algorithms that control consumption patterns of media content. Echo chambers lead to movement within homogeneous information bubbles on social media [16]. This is most likely to instigate and perpetuate conspiracy beliefs. High exposure to conspiracy-theoretical content contributes to its dissemination and acquirement (cf. [17,18,19]).

Currently more than four billion people, and thus more than half of the world’s population, are using social media [20,21]. In addition, the importance of social media as a source of news and information has increased, according to recent statistics, especially due to the COVID-19 pandemic [21]. Both science-based media news and fake news spread quickly and easily on social media [15]. Facebook seems to be the main channel for spreading false information related to conspiracy theories [21]. 

## 2. Theoretical Background

### 2.1. Affirmation of Meaning Frameworks and the Big-Cause Effect

People are meaning-makers who capture the external world by their mental representations. If their mental representation of the external world is disrupted by contradictory evidence, people experience a deep concern with the unexpected incongruity, which activates their meaning maintenance system, motivating them to employ alternative frameworks of meaning [22].

The emergence of the COVID-19 pandemic threatens the existing framework of meaning because it contradicts the predictability of events and causes a loss of control, which enhances the responsiveness to conspiracy theories [23]. The pandemic represents a global change that has overwhelming consequences for many people and disrupts their belief system. In correspondence with the big-cause effect a readiness to explain big events by big causes is triggered. For example, the occurrence of devastating damage triggers the tendency to apply a big cause more than the occurrence of less devastating damage [23]. Therefore, the COVID-19 pandemic is likely to be explained by conspiracy theories in accordance with the meaning maintenance system and the big-cause effect. In summary, loss of control, feelings of uncertainty because of disruption of belief systems, and threat to the social order, which accompanies the emergence of the COVID-19 pandemic facilitate the employment of conspiracy theories, which offer an alternative meaning after threatening the predominant meaning maintenance system. The occurrence of the big-cause effect does the rest. As a consequence, belief in conspiracy theories is likely to be intensified.

### 2.2. Beliefs and Attitudes as Tools of the Meaning-Maker 

Meanings are carried by beliefs and attitudes. In the following, we focus on beliefs. Beliefs represent the cognitive basis of attitudes that include an additional affective component of liking or disliking (cf. [24,25]). The opinions on ideas and ideologies are reflected in beliefs about them [26,27,28,29,30,31]. Beliefs and attitudes that are derived from them serve as frameworks of personal meaning. Therefore, beliefs and attitudes toward COVID-19-related media content serve as building blocks for the attribution of meaning to this media content.

The concept of persuasion describes how social influence occurs (cf. [32]). It is defined as attitude change in response to verbal messages [33]. Early research on persuasion was reported by Hovland and his co-workers at Yale University [32]. Petty and Cacioppo [34] in their Elaboration Likelihood Model, as well as Eagly and Chaiken [24] in their Heuristic-Systematic Model, proposed influential dual process models of persuasion, which were widely applied in recent research on attitude change. Both models contrast a fast route of attitude change with a more time-consuming route, which is based on the systematic elaboration of arguments.

Both models have much in common, but the Heuristic-Systematic Model emphasizes the use of heuristics more than the Elaboration Likelihood Model. Basically, the ELM distinguishes between two routes of persuasion processes depending on the likelihood of elaboration. A distinction is made between the central route, which requires systematic information processing, and the peripheral route, which is based on heuristics and the availability of superficial cues. On the central route, a critical weighing of argumentative content takes place by investing high cognitive effort. This is referred to as high elaboration likelihood. The peripheral route stands for the processing of heuristic cues as cognitive shortcuts neglecting the quality of arguments and requiring little cognitive effort. 

A review of prior research suggests that three basic motives motivate belief in conspiracy theories: the epistemic motive, the existential motive, and the social motive [5]. In this research, the focus is on the epistemic motive, which explains conspiracy beliefs by feelings of uncertainty that are likely to be aroused by the COVID-19 crisis [1,2,5] and which contrast with a desire for certainty. The epistemic motive represents the process of sense making [22]. Conspiracy beliefs fulfill the sense-making function because they answer questions such as ‘who is responsible?’ and ‘which factors cause the threat to the social order?’.

The acceptance of conspiracy beliefs is determined in several ways. Specifically, van Prooijen et al. [35] suggested a distinction between negative and neutral/positive determinants of acceptance of conspiracy beliefs. Whereas right-wing orientation, anxiety, sense of uncontrollability, and affective instability constitute the negative determinants, NFC, openness to experience, need for uniqueness, and sensation seeking represent the neutral/positive determinants. These neutral/positive determinants refer to openness to the unexpected, a preference for the less popular approach to an issue, a propensity to delve into a subject, and a preference for thinking through complex issues. The common denominator seems to be the need for epistemic curiosity (cf., [36]). This assumption corresponds with the epistemic motive (see above).

### 2.3. Conspiracy Theories in Social Media

#### 2.3.1. Scientific Theory vs. Conspiracy Theory

In general, scientific and conspiracy-theoretical media content can be distinguished, which oppose each other on a continuum of verifiability [16]. Conspiracy-theoretical content is likely to elicit less systematic elaboration of arguments. In correspondence with this assumption conspiracy media content are related to emotional rather than analytical processing [5]. Scientific media content is mostly characterized by arguments based on empirical sources [18]. Conspiracy-theoretical media content, on the contrary, presents explanations lacking an empirical basis. Moreover, conspiracy theories use often unsubstantiated assumptions as explanations of public events and societal actions [37]. Recent studies show that conspiracy beliefs are increasingly spreading in the context of COVID-19 on social media [2,38,39]. However, it is not clear which factors influence belief toward scientific versus conspiracy-theoretical postings. On the basis of this reasoning the following research question was derived.

**RQ1:** Is the belief in the scientific posting higher than the belief in the conspiracy-theoretical posting?

#### 2.3.2. Need for Cognition 

In contrast conspiracy-theoretical content is likely to elicit less systematic elaboration of arguments. In correspondence with this assumption conspiracy media content are related to emotional rather than analytical processing [5].

According to the ELM, the probability of elaboration depends on the processing motivation and ability, respectively [34]. The higher the processing motivation/ability, the more likely systematic elaboration of arguments occurs. Processing motivation/ability can be measured by an individual’s need for cognition (NFC) which represents individual differences in how much and how readily the person thinks about the arguments contained in a message [40]. According to the theoretical framework of NFC, high NFC should enhance the persuasive impact of high-quality arguments. From the theoretical framework of NFC H1 was derived.

**H1:** High NFC increases the belief in scientific theories about COVID-19 related postings, whereas it decreases the belief in corresponding conspiracy theories.

#### 2.3.3. Conspiracy-Theoretical Worldview

Studies show that the belief in conspiracy theories is based on increased exposure to such media content [17,18,19]. Moreover, social media trigger the spread of conspiracy-theoretical misinformation more than face-to-face communication (cf. [15]). Research indicates that a conspiracy-theoretical worldview underlies the beliefs of conspiracists [6,26,27,28,29,30]. This assumption was also confirmed in the context of COVID-19 related CTs [2,38]. 

However, the meaning of social media and the comparison to scientific news in this context remains unclear. According to the review by Wang et al. [41], social media particularly contributes to the spread of health-related conspiracy-theoretical misinformation. This was shown by a negative association between conspiracy-theoretical COVID-19 related media content and the acceptance of government health policies [38,42]. In accordance with this reasoning, H2 was formulated.

**H2:** A conspiracy-theoretical worldview decreases the belief in scientific COVID-19 related postings, while it increases the belief in conspiracy-theoretical postings [2,29,30].

#### 2.3.4. Big 5 Variables

Much personality research is guided by the five-factor (Big 5) model of personality by McCrae and Costa [43], which in general has been shown to be robust and valid. Two of the five factors (agreeableness and openness to experience) have been connected with conspiracy beliefs [44]. Agreeableness is defined as a willingness to defer to others during interpersonal conflict ([45], p. 143) including trust and forgiving attitude ([43], whereas openness to experience is defined as a need for variety, novelty, and change ([45], p. 143) including original and imaginative thinking, broad interests and daring [43] One might argue that agreeableness is related to conformity. As a result, high agreeableness might lead to a readiness to agree with conspiracy theories that are supported by a majority. Furthermore, original and imaginative thinking might be activated by conspiracy theories that emphasize secrets and mysteries. However, previous research revealed inconsistent findings on the relationship between openness to experience and agreeableness on the one hand and conspiracy beliefs on the other hand [27,28,29,30,46,47,48]. Inconsistent findings are also reported with respect to the association between neuroticism and belief in conspiracy theories, although a comprehensive study [46] revealed a significant correlation indicating a positive relationship. Neuroticism is defined as a tendency to experience dysphoric affect including low self-esteem and pessimism ([45], p. 143).

In correspondence with the inconsistent results, a recent meta-analysis by Goreis and Voracek [44] indicated that the average correlations between agreeableness, openness to experience, and neuroticism on the one hand and conspiracy beliefs on the other hand were close to zero. Because the results were heterogeneous across samples moderator analyses were conducted indicating with respect to openness to experience that samples with larger proportions of men and samples consisting of younger participants exhibited higher correlations of openness to experience with conspiracy beliefs. In addition, the association between agreeableness and conspiracy beliefs was higher among samples that included a larger proportion of older participants.

In summary, the results on the association between openness to experience/ agreeableness and belief in conspiracy theories are inconsistent. Therefore, the following research questions were formulated.

**RQ2:** How are the personality traits (a) agreeableness and (b) openness to experience associated with belief in scientific or conspiracy theories embedded in COVID-19-related media content?

The structure of hypotheses and research questions is illustrated in Figure 1.

## 3. Method

### 3.1. Study Design

The survey consisted of an online questionnaire (The study and its aims were preregistered: https://doi.org/10.17605/OSF.IO/9BSGX). Data collection spanned a 5-week period, starting on 12 March 2021 and ending on 15 April 2021. The survey of the preliminary study lasted from 5 to 11 March 2021. Participants took a maximum of 15 min to complete the survey. The central variables to be collected in this study were the belief in scientific postings and in conspiracy postings. In addition, NFC, conspiracy-theoretical worldview, and the Big 5 personality traits of agreeableness and openness to experience were measured. The inclusion criterion was the daily use of Facebook (Do you log in on Facebook at least once per day? 1 = Yes, 2 = No). 

### 3.2. Materials

#### 3.2.1. Preliminary Study of Media Content

The aim of the preliminary study was to the selection of the media content that was operationalized by two fictitious Facebook postings. The focus is on Facebook postings because Facebook is considered a key medium in spreading fake news [49]. The content of the postings dealt with the effectiveness and mandatory use of FFP masks [50]. The FFP mask requirement represents an example of a COVID-19-related topic that elicits conflicting opinions in society. 

In the preliminary study, the participants read a total of six fictitious COVID-19-related postings that differed in their argument quality and source credibility. A within-subjects design was employed, including six postings. Three fictitious scientific (see Appendix A Figure A1, Figure A2 and Figure A3) and conspiracy-theoretical (see Appendix A Figure A4, Figure A5 and Figure A6) COVID-19-related postings were included. The construction of the postings assumed that high argument quality and high source credibility are characteristics of a scientific posting [5,16]. 

In contrast, these same variables are assumed to be low for conspiracy-theoretical postings. Participants rated the six postings on two items, source credibility of the postings on a scale from 1 (*not at all credible*) to 5 (*very credible*) and argument quality of the postings from 1 (*very weak*) to 5 (*very strong*).

The sample of the preliminary study included predominantly young adults with a mean age of 29 years (*SD* = 13 years). The 71% of the 90 participants were female. The highest level of education attained by most participants was Abitur (German high school diploma, (49%) or university degree (40%). In addition, most of the participants were currently studying (57%) or working (34%). 

Table A1 (see Appendix B) summarizes the mean ratings of credibility and argument quality of the six postings. For the main study, scientific media content was represented by Posting Team A (see Appendix A Figure A1), exhibiting both the highest mean argument quality (*M* = 4.20, *SD* = 0.82) and source credibility (*M* = 4.10, *SD* = 0.79). Due to a lack of normal distribution by a Shapiro–Wilk test (*W*s ≥ 0.74, *p* < 0.001), we used the nonparametric Wilcoxon test to compare means. Regarding both assessments, Posting Team A differed significantly from both scientific Posting Team D and Posting Team E (see Appendix B Table A2). Posting Team B represented the conspiracy-theoretical posting in the main study (see Appendix A Figure A4). Posting Team B achieved the lowest source credibility (*M* = 1.63, *SD* = 0.73), which was significantly different from Posting Team C (*z* = −2.22, *p* = 0.026, *r* = 0.44) and Posting Team F (*z* = −2.15, *p* = 0.031, *r* = 0.29). In addition, Posting Team B’s argument quality (*M* = 1.79, *SD* = 0.73) was, on average, assessed low. In summary, based on the results of the preliminary study, Posting Team A served as the scientific media content and Posting Team B as the conspiracy-theoretical media content in the main study. 

#### 3.2.2. Belief in Conspiracy Theories

Based on previous research [6,26,27,28,29,30,31] belief in conspiracy theories and scientific theories, respectively, was measured by four items (*How credible/believable/plausible/convincing do you find the posting?*). Responses were obtained on a 7-point Likert scale (1 = *strongly disagree* to 7 = *strongly agree*). The dependent variable was the sum across the four items for CT and ST, respectively. Based on the current sample, the internal consistency of both scales was very high (α _CT_ = 0.95 and α _SCI_ = 0.96). 

Additionally, we conducted a confirmatory factor analysis of the ratings of CT and ST using MPlus 8.6 [51]. We used the mean and variance adjusted unweighted least squares method was used (ULSMV), that has to be shown as default estimator for models containing ordinal outcomes. Above, the ULSMV was proven to be robust regarding model violations [52]. The model fit was assessed using four statistics: (a) the chi-square test statistic, (b) the Comparative Fit Index (CFI; an acceptable fit is inferred if the CFI is 0.90 or higher), (c) the Tucker–Lewis index (TLI; an acceptable fit is inferred if the TLI is 0.90 or higher) and (d) the Root Mean Square Error of Approximation (RMSEA; an acceptable fit is inferred if the RMSEA is equal to 0.08 or smaller). The CFA revealed a suitable fit: chi-square, *p* < 0.05, CFI = 0.996, TLI = 0.995, RMSEA = 0.058. Therefore, the CT items were located on the first factor and the ST items on the second factor. 

#### 3.2.3. Need for Cognition

Need for Cognition was measured by the questionnaire originally developed by Cacioppo and Petty (1982). The German NFC-short scale [53] consists of only four items (e.g., *I would prefer complex to simple problems*.). The response scale of the *NFC-short scale* ranges from 1 (*strongly disagree*) to 7 (*strongly agree*). Beißert and colleagues [53] stated that the calculation of internal consistencies as a reliability measure for the NFC-short scale is not appropriate because of its shortness. In correspondence with the findings [53], the internal consistency of the short scale in the current sample was low (α = 0.42). An alternative is the assessment of the retest reliability of the scale. Using the test-retest method, Beißert and colleagues [53] obtained sufficiently high reliability coefficient of the short scale (*r_tt_* = 0.78). 

#### 3.2.4. Conspiracy-Theoretical Worldview

The conspiracy-theoretical worldview was assessed using the German version of the *Conspiracy Mentality Questionnaire* (*CMQ,* 27). The *CMQ* asks participants to assess their tendency toward a conspiracy-theoretical worldview on five items (e.g., *I think government agencies closely monitor all citizens*.) employing an 11-point response scale (0*% certainly not,* 10*% extremely unlikely,...,* 100*% certainly).* Bruder et al. consider the *CMQ* to be reliable (α = 0.82). The reliability estimate of the CMQ was also high in the current sample (α = 0.88).

#### 3.2.5. Agreeableness

The Big 5 personality trait agreeableness was measured by six items of the respective subscale of the 30-item short version of the German *NEO Five-Factor Inventory* (*NEO-FFI-30*; [54]). The positive pole of agreeableness refers to trust, cooperation, and politeness. The negative pole of antagonism includes facets of arrogance, aggressiveness, and manipulativeness [43]. A sample item is *Some people think I am cold and calculating.* The response scale ranges from 1 (*strongly disagree*) to 5 (*strongly agree*). The agreeableness scale revealed sufficient internal consistency in the current sample (α = 0.69). Comparable results were reported by Körner et al. (α = 0.75).

#### 3.2.6. Openness to Experience

Openness to experience includes receptiveness to new ideas, values, and feelings, originality, imaginativeness, and broad interests [43]. It was measured by 6 items of the respective subscale of the 30-item short version of the German *NEO Five-Factor Inventory* (*NEO-FFI-30*; [54]). A sample item is *Poetry impresses me little or not at all*. The response scale ranges from 1 (*strongly disagree*) to 5 (*strongly agree*). The scale revealed sufficient reliability both in the current sample (α = 0.72) and in the calibration sample (α = 0.67). 

#### 3.2.7. Demographic Variables

We included the following demographic variables: sex (female, male, diverse), age (in years), highest educational status (no graduation, German “Hauptschulabschluss”, German “Mittlere Reife”, German “Fachabitur”, German “Abitur”, completed an apprenticeship, completed university studies, other), current occupation (homemaker, in apprenticeship, studying, in employment, retired, unemployed, other), and marital status (single, in a relationship, married, separated, divorced, widowed) were included in the questionnaire. Furthermore, we applied quality controls for data collection integrating the item. To check data quality, please indicate the answer option strongly disagree“ twice, for example, as part of the agreeableness subscale. Data of participants who did not select the requested answer were excluded.

### 3.3. Sample

Using the snowball sampling technique, we distributed the invitation link for the online questionnaire via social media (WhatsApp, Telegram, Facebook), including information flyers about the study. We explicitly recruited participants in Telegram groups of the German movement “Querdenken” (i.e., a German protest movement including pandemic skeptics, anti-vaxxers, and anti-lockdown protesters). This movement opposes anti-Corona measures of the German government and is open-minded about conspiracy theories. Participant recruitment occurred via a study portal of the Ruhr University Bochum. Overall, the sample was quite young, representing a broad demographic background of participants.

According to a power analysis by G*Power 3.1 [55], the optimal sample size for an assumed medium effect (*F*^2^ = 0.15) was 89 participants. In total, we recruited 274 participants. Data set cleaning led to the exclusion of 95 participants because of minority status refusing to consent to participate, failed quality control, and incomplete data. In addition, four additional participants were eliminated from the sample because of extreme scores.

After data set cleaning, the final sample consisted of 175 participants, 118 participants of whom were female, 56 participants male, and 1 participant diverse. We acknowledge that the recruited sample exceeded the minimally needed sample size as indicated by a priori analysis. This, however, was done to increase the precision of our statistical model. The mean age was 29 years (*SD* = 12 years). Half of the participants had a high school diploma (German “Abitur”, 54%) as their highest educational qualification, and one-third had a university degree (30%). Most of the participants were students (59%) or employees (34%). 41% of the participants were in a romantic relationship without marriage, 38% were single, 19% were married, and ≤1% were divorced or separated.

### 3.4. Statistical Analysis

In the first step, descriptive analyses were performed. Since the independent variable in this work was not based on a normal distribution according to Kolmogorov–Smirnov test and Shapiro–Wilk test, we used robust methods for the main hypothesis-based analyses. The computations of the model assumptions can be understood in the provided data sets and syntax. For testing RQ1, nonparametric Wilcoxon tests, which test differences between two dependent samples as nonparametric equivalent to the *t*-test for dependent samples, were employed to compare the belief in perceived scientific and conspiracy-theoretical media content. In addition, linear regression analysis is to examine further hypotheses and research questions. All models were controlled for sex, age, highest educational status, current occupation, and marital status, which were included as covariates. Finally, the tested significance of differences between correlations in magnitude was examined by Fisher’s *z*-test. All statistical analyses were performed using IBM SPSS 27 statistical software [56] (the data set is available at https://osf.io/twnbz/?view_only=77d2c71c3aba4299b28b65155f063c66).

## 4. Results

### 4.1. Descriptive Analysis

The variables NFC, conspiracy-theoretical worldview, agreeableness, and openness to experience were approximately normally distributed, which was shown in the visualization of the distribution and skewness (vs. ≤|0.45|. Further descriptive statistics are summarized in Table 1. Correlational results indicated, as expected, that a significant negative association between belief in scientific theories and belief in conspiracy theories occurred, explaining 47.6% of the total variance (*r* = −0.69, *p* < 0.001). Overall, 155 participants (89%) indicated that their belief in the scientific theory was stronger than their belief in the conspiracy theory. 

### 4.2. Research Questions

With respect to RQ1, belief in the scientific posting (*M* = 5.51, *SD* = 1.37) was significantly higher than belief in the conspiracy-theoretical posting (*M* = 2.55, *SD* = 1.61), *z* = −9.20, *p* < 0.001, *r* = −0.69. Note that we employed the nonparametric Wilcoxon test to examine the hypothesis because a Shapiro–Wilk test revealed a violation of the assumption of normal distribution for both beliefs (*W* = 0.85, *p* < 0.001). 

The research question 2 refers to the association between Big 5 variables and belief in scientific and conspiracy postings. Specifically, agreeableness and openness to experience were included. Results indicate that agreeableness was neither related to belief in the scientific posting nor to belief in the conspiracy posting. Agreeableness did neither affect the belief in scientific theories (all *p*s ≥ 0.061) nor the belief in conspiracy theories (all *p*s ≥ 0.324), even after taking into account control variables. However, agreeableness and conspiracy-theoretical worldview were associated negatively, *r* = −0.22, *p* = 0.004, *R²* = 0.048.

Finally, openness to experience was positively associated with belief in the conspiracy posting, *F*(1,173) = 4.27, *p* = 0.040, *R*² = 0.024, but did not predict belief in the scientific posting. While the effect for the conspiracy posting remained significant after controlling for sex (*p* = 0.028) and age (*p* = 0.048), it was no longer significant after controlling for highest educational status, current occupation, and marital status (all *p*s ≥ 0.071). The effect for the scientific posting remained nonsignificant even after taking into account control variables (all *p*s ≥ 0.630). The correlations of openness to experience with both belief variables did not differ significantly, *t*(174) = −1.32, *p* = 0.095.

### 4.3. Hypotheses Tests

Linear regression analysis including bootstrapping was employed to examine H1 and H2. Table 2 summarizes the prediction of belief in the scientific posting by NFC, a conspiracy-theoretical worldview, agreeableness, and openness to experience. The results of the respective regression analysis with belief in conspiracy theories as an outcome variable are summarized in Table 3. 

H1, which focused on the need for cognition, was not confirmed because NFC did not predict the belief in the scientific posting. Even after controlling for the effects of age, sex, highest educational status, current occupation, and marital status, no significant effect of NFC emerged (all *p*s ≥ 0.436). In contrast, NFC predicted the belief in the conspiracy posting, *F*(1,173) = 7.47, *p* = 0.007, *R*² = 0.041. The magnitude of the correlations between NFC and belief in the scientific posting (*r* = −0.06) and NFC and belief in the conspiracy posting (*r* = 0.20), respectively, was significantly different, *t*(174) = 1.92, *p* = 0.028, indicating the occurrence of appreciable differences in the magnitude of correlations. Thus, the higher the NFC, the higher the belief in conspiracy, whereas NFC and belief in scientific theories were unrelated. This effect remained significant even when the control variables were held constant (all *p*s ≤ 0.027). 

In correspondence with H2, the conspiracy-theoretical worldview negatively predicted belief in the scientific posting, *F*(1,173) = 42.10, *p* < 0.001, *R*^2^ = 0.196, and positively predicted belief in the conspiracy posting, *F*(1,173) = 74.31, *p* <0.001, *R*^2^ = 0.300. Even after statistically removing the effects of control variables, the conspiracy-theoretical worldview was negatively linked with the scientific posting (*p* < 0.001). When the control variables were included in the regression model, the influence of conspiracy-theoretical worldview on belief in conspiracy theory remained significant (*p* < 0.001). The magnitude of the correlations between a conspiracy-theoretical worldview, on the one hand, and the belief in scientific postings and the belief in conspiracy postings, respectively, on the other hand, differed significantly, *t*(174) = 8.49, *p* < 0.001. 

In summary, the stronger the conspiracy-theoretical worldview, the weaker the belief in scientific theories and especially the stronger the belief in conspiracy theories. 

## 5. Discussion

The main purpose of the study was to examine determinants of the belief in conspiracy-theoretical media content contrasted with the belief in scientifically grounded media content. The focus was on belief in conspiracy postings with respect to COVID-19-related media content.

The results indicate that belief in the scientific posting is higher than belief in the conspiracy-theoretical posting. In general, the level of belief in a particular posting is quite specific to the content of the posting itself. Nevertheless, the result corresponds with the reasoning that the higher argument quality of the scientific posting in comparison with the conspiracy posting was crucial. Note that the preliminary study confirmed that the argument quality of the scientific posting was higher than the argument quality of the conspiracy posting. 

Contrary to H1, NFC enhanced the belief in conspiracy-theoretical media content. Furthermore, NFC did not affect the belief in the scientific posting. On the surface, these results seem to contradict the reasoning by Petty and Cacioppo [34] which seems to imply that high NFC enhances the critical weighing of arguments on the cognitive level. The results of the preliminary study indicate that argument quality was higher for the scientific compared to the conspiracy-theoretical posting. Other studies suggest, in accordance with the viewpoint by Petty and Cacioppo [34], a positive relationship between conspiracy-theoretical views and low cognitive reflection [57] and high intuition instead of analytical reasoning [3].

NFC represents the epistemic motive that explains conspiracy beliefs through feelings of uncertainty (cf., introduction). The frame of reference of NFC overlaps with other constructs that are related to the acceptance of conspiracy beliefs and fit into the framework of the epistemic motive. One such construct that was included in our study is openness to experience. Remarkably, NFC and openness to experience both foster the belief in the conspiracy-theoretical posting (cf., Table 1). Furthermore, NFC and openness to experience correlate significantly positively, r = 0.37, *p* < 0.01, indicating 14% of the common variance between both constructs. Note that previous research by Berzonsky and Sullivan [58] has already found this result. Two other variables—sensation seeking and the need for uniqueness—fit into the framework of the epistemic motive and are positively related to conspiracy beliefs [35,59]. Sensation seeking is linked to a heightened interest in entertainment. “Conspiracy theories typically involve spectacular narratives that include mystery, suspected danger, and unknown forces that one does not fully comprehend” ([35], p. 26). These features presumably are especially attractive to high sensation seekers comparable to their fascination with horror movies.

Openness to experience reflects a need for variety, novelty, and change [45], which includes an interest in understanding and pursuing new perspectives. Conspiracy theories tend to focus on such new perspectives. High openness to experience is likely to facilitate an interest in unusual and unique ideas. Conspiracy theories frequently incorporate unusual and unique ideas. Therefore, the positive association between openness to experience and belief in conspiracy postings corresponds with the focus on new perspectives inherent in openness to experience. The association between openness to experience and belief in conspiracy theories is modified by sample characteristics such as the proportion of women in the sample and the age of participants [44]. Nevertheless, the emergence of a reliable positive association between openness to experience and belief in conspiracy theories corresponds with results for NFC, sensation seeking, and the need for uniqueness. Furthermore, conspiracy beliefs are associated with entertainment facets that could also satisfy the need for curiosity [35].

The personality trait of openness to new experiences is also associated with unconventionality and liberal political attitudes [60]. In this respect, we find parallels to the non-acceptance of generally valid preventive measures such as mask-wearing [1,11,12,13,14]. 

Because NFC overlaps considerably with openness to experience, the same reasoning might be applied to the positive association between NFC and belief in the conspiracy posting. The content of openness to experience, which overlaps with NFC, seems to enable people who express high NFC to approach conspiracy postings with positive interest, although the component of analytical reasoning seems to favor the belief in the scientific posting. Interestingly, Cacioppo et al. (1996) speculate about two facets of NFC, referring to the positive association between openness to experience and NFC. One facet describes individual enjoyment of effortful cognitive engagement, whereas another facet represents a positive self-rating of cognitive abilities [40].

The explanation of the positive association between NFC and belief in the conspiracy posting depends both on the component of openness to experience, which seems to be inherent in NFC, and the assumption that the facet of openness to experience implied by NFC has a stronger effect on the evaluation of the conspiracy posting than the emphasis on analytical reasoning implied by NFC, which is likely to reduce the belief in the conspiracy posting. Further research is needed to clarify these issues. However, an advantage of the proposed explanation for the positive association between NFC and conspiracy belief is that it is able to account for the positive association of both NFC and openness to experience with conspiracy belief and the positive association between NFC and openness to experience. 

H2 proposes that a conspiracy-theoretical worldview decreases the belief in the scientific COVID-19-related posting, whereas it increases the belief in the conspiracy posting. The results correspond with H3 because they indicate that a conspiracy-theoretical worldview increases the belief in the conspiracy-theoretical media content, whereas it decreases the acceptance of the scientific media content. 

In addition, the association between worldview and belief in the conspiracy posting is stronger than the association between worldview and belief in the scientific posting, indicating that a similarity effect occurs, meaning that the conspiracy-theoretical worldview fits into the content of the conspiracy posting. The conspiracy-theoretical worldview also inhibits the belief in the scientific posting, but the similarity effect is stronger than the inhibition effect. 

Therefore, the conspiracy-theoretical worldview strongly facilitates the belief in the conspiracy posting and somewhat inhibits the belief in the scientific posting. This pattern of results indicates that a conspiracy-theoretical worldview tends to generalize strongly on a similarity gradient, whereas the inhibition of the belief in the scientific posting is somewhat weaker. In general, these results correspond with prior research on the association between conspiracy-theoretical worldview and belief in conspiracy-theoretical media content [2,6,26,27,28,29,30,38]. 

## 6. Limitations and Outlook

In summary, the following factors were positively associated with the belief in the COVID-19 conspiracy-theoretical posting: conspiracy-theoretical worldview, NFC, openness to experience, and age. In addition, males scored higher than females.

The present study has some limitations: First, the sample composition did not achieve as much heterogeneity as expected. Despite a large sample size, the representativeness of the sample is limited because younger participants and women are overrepresented. In addition, data were collected by self-report, which may be biased by social desirability and other response sets. 

In general, research on conspiracy beliefs relies heavily on self-reports elicited by verbal items. Nevertheless, future studies might employ alternative measurements of belief. For example, nonreactive measures and methods of indirect measurement (cf., [61]) might complement verbal measures. Specifically, categorization tasks are employed, which enable the indirect measurement of beliefs via reaction times. (cf., the Implicit Association Test (IAT, [61]). Such spontaneous, automatic responses reduce the conscious distortion of responses.

Furthermore, conspiracy belief and scientific belief, respectively, were measured by the assessment of single postings. Nevertheless, the reliability of the belief measures was high. To increase the generalizability of findings across stimulus materials, future studies should include manifold postings. 

Finally, the selection of possible determinants of conspiracy beliefs was limited. For example, newly introduced determinants such as sensation seeking and the need for uniqueness should be taken into account in future research. In addition, recent research [46] indicates that neuroticism is defined as a tendency to experience dysphoric affect ([45], p. 143), which might increase conspiracy beliefs. Because neuroticism is a marker of uncertainty and because conspiracy theories offer a framework of meaning, they might be especially attractive to anxious people. It would be desirable to include personality variables from the complete five-factor model of personality in future studies in order to take into account the most important basic dimensions of personality.

The dissemination of conspiracy theories constitutes a social problem in the context of the containment of the COVID-19 pandemic. Previous research demonstrated a negative association between conspiracy-theoretical beliefs and the acceptance of empirically based preventive measures for pandemic containment [1,11,12,13]. In addition, Romer and colleagues [14] demonstrated a negative association between conspiracy beliefs and vaccination propensity with respect to COVID-19. 

This implies the need to limit the influence of conspiracy-theoretical postings. Fake news spreads faster and broader in social media than the truth [62]. People are meaning seekers [22] who focus on alternative theories if their meaning maintenance system is called into question. Therefore, it is urgent to support the acceptance of science-based news as well as the acceptance of preventive measures for COVID-19. Such an endeavor would profit from including the central route of persuasion proposed by the ELM. 

Another way to improve the acceptance of scientific content among conspiracists would be the dissemination of short and understandable explanatory videos. During the COVID-19 pandemic, the use of such videos was already demonstrated in Germany by May Thi Nguyen-Kim, a science journalist who was awarded a Federal Cross of Merit [63]. Whether such videos provide convincing arguments instead of conspiracy-theoretical arguments could be tested by follow-up studies. 

In general, scientific and conspiracy narratives about the causes of COVID-19 compete on social media [18]. The present study offers new insights into the message characteristics that are likely to make sure that the scientific narrative surpasses the conspiracy narrative about COVID-19 in social media by identifying factors that determine conspiracy-theoretical beliefs and informing about applied implications. The results enable the derivation of measures of social action against the current COVID-19 pandemic. For example, an opinion attack on conspiracy-theoretical worldviews seems to be promising. Whereas such an attack is located on the ideological level, additional anti-conspiracy communications that are promising are located on the individual level. 

The ELM by Petty and Cacioppo [34] considers the relevance of argument quality in persuasive communications and distinguishes between two routes of persuasion depending on the likelihood of elaboration. The central route, which takes the quality of arguments into account, is contrasted with the peripheral route, which puts less emphasis on argument quality. The central route requires a careful weighing of argumentative content. 

The findings of the present study indicate that both NFC and openness to experience are positively associated with a high probability of elaboration in terms of the ELM. Because both determinants are positively related to systematic cognitive processing and preference for novel viewpoints, in a first step, it is necessary to revise the stereotype of conspiracists as superficial people. Although many conspiracists are characterized by right-wing orientation, affective instability, and loss of control, others are attracted by the promise of entertainment. Our general idea is that successful campaigns might focus on specific subgroups of people. Campaigns might be designed especially for target persons who score high on sensation seeking and openness to experience. For example, a campaign against the acceptance of conspiracy theories could mention the entertainment value of conspiracy theories for many readers and refer (more or less ironically) to possible replacements for reading conspiracy theories in terms of their promise of entertainment, such as, for example, watching scary movies on Netflix. Such a campaign could align the stimulus value of conspiracy theories and scary movies (or other emotion-arousing movies) and conclude that the replacements are the better choice. This is only one example of how to frame an anti-conspiracy message, which addresses people who exhibit a high need for epistemic curiosity. In general, campaigns that emphasize alternatives to delving into conspiracy theories seem to be promising.

In addition, a subgroup of people who are interested in conspiracy theories and score high on the need for cognition are likely to focus on the central route of information processing, preferring an argumentative discussion, which allows the refutation of conspiracy theories in detail and includes counterarguments. Such an approach might turn out to be a successful campaign against misconceptions incorporated in conspiracy theories. Emphasis on new perspectives that are derived from scientific thinking is likely to be especially convincing for people who exhibit a strong need for cognition.

## Figures and Tables

**Figure 1 behavsci-12-00435-f001:**
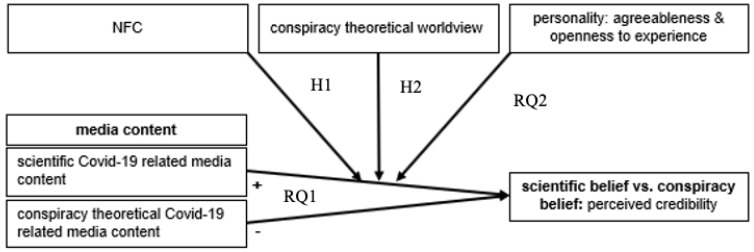
Structure of hypotheses and research questions.

**Table 1 behavsci-12-00435-t001:** Means, standard deviations, and correlations.

Variable	*M*	*SD*	1	2	3	4	5	6
Criteria:								
1. Belief in scientific theories	5.51	1.37	−					
2. Belief in conspiracy theories	2.55	1.61	−0.69 **	−				
Moderators:								
3. Need for cognition	4.65	0.73	−0.06	0.20 **	−			
4. Conspiracy-theoretical worldview	5.30	2.10	−0.44 **	0.55 **	0.05	−		
5. Openness to experience	3.61	0.66	−0.03	0.16 *	0.37 **	−0.00	−	
6. Agreeableness	3.94	0.58	0.14	−0.08	0.07	−0.22 **	0.15 *	−

*Note. M* = mean, *SD* = standard deviation. *df* = 173.* *p* < 0.05. ** *p* < 0.01.

**Table 2 behavsci-12-00435-t002:** Regression analysis: belief in scientific theories as criterion.

Model ^a^		*ß*	*SE*	BCa 95% CI ^b^		*t*	*p*	*R²*
				LL	UL			
1	(Intercept)		0.67	4.80	7.19	8.98	<0.001	
	Need for cognition	−0.06	0.14	−0.38	0.17	−0.78	0.436	0.004
2	(Intercept)		0.25	6.50	7.56	27.70	<0.001	
	Conspiracy-theoretical worldview	−0.44	0.05	−0.39	−0.18	−6.49	<0.001	0.196
3	(Intercept)		0.71	2.93	5.42	5.92	<0.001	
	Agreeableness	0.14	0.18	0.004	0.67	1.89	0.061	0.020
4	(Intercept)		0.58	4.68	6.74	9.85	<0.001	
	Openness to experience	−0.03	0.16	−0.36	0.25	−0.35	0.729	0.001

*Note.* CI = confidence interval; LL = lower limit; UL = upper limit. *df* = 174. ^a^ demographic variables had no significant effect on the statistical model, all *p* >0.05. ^b^ Confidence interval and standard deviation via BCa-Bootstrapping with 10,000 BCa-samples.

**Table 3 behavsci-12-00435-t003:** Regression analysis: perceived credibility of the conspiracy-theoretical posting as criterion.

Model ^a^		*ß*	*SE*	BCa 95% CI ^b^		*t*	*p*	*R²*
				LL	UL			
1	(Intercept)		0.77	−1.12	1.99	0.61	0.543	
	Need for cognition	0.20	0.16	0.12	0.80	2.73	0.007	0.041
2	(Intercept)		0.28	−0.18	0.88	1.18	0.241	
	Conspiracy-theoretical worldview	0.55	0.05	0.30	0.53	8.62	<0.001	0.300
3	(Intercept)		0.84	1.85	4.91	4.04	<0.001	
	Agreeableness	−0.08	0.21	−0.58	0.17	−0.99	0.324	0.006
4	(Intercept)		0.67	−0.13	2.51	1.77	0.078	
	Openness to experience	0.16	0.18	0.01	0.75	2.07	0.040	0.024

*Note.* CI = confidence interval; LL = lower limit; UL = upper limit. *df* = 174. ^a^ demographic variables had no significant effect on the statistical model, all *p* >0.05. ^b^ Confidence interval and standard deviation via BCa-Bootstrapping with 10,000 BCa-samples.

## Data Availability

The data sets generated and analyzed during the current study are available in the OSF repository, https://osf.io/twnbz/?view_only=77d2c71c3aba4299b28b65155f063c66.

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
