# Peer review of "Science vs. Conspiracy Theory about COVID-19: Need for Cognition and Openness to Experience Increased Belief in Conspiracy-Theoretical Postings on Social Media"

_behavsci, 2022, doi:10.3390/bs12110435_

Round 1
Reviewer 1 Report
The manuscript entitled “Science vs. Conspiracy-Theory about Covid-19: Need for cognition and openness to experience increase belief in conspiracy- theoretical postings in social media” addresses a very actual and interesting topic – conspiracy theories beliefs – by focusing on its psychological determinants.
I appreciated the clarity and the systematicity of the manuscript. However, I have some comments.
- Measures: I think that the Authors should better justify their choice to consider only two personality traits. Even though, Openness and Agreeableness have been more frequently found to be associated with Conspiracy beliefs, literature suggests, for example, that also Neuroticism could be correlated to Conspiracy beliefs (i.e. to reduce uncertainty, Hollander, 2018). Please, expand more on that.
- Procedure: please clarify when data collection exactly took place. I think it should be relevant to include this piece of information (and maybe it would be useful to discuss it)
- How familiar were participants with Facebook? How did they rate Facebook reliability? Did you collect these information?
- Discussion: I think that the discussion concerning H2 needs further clarification. Being a counterintuitive result, it should be better justified
- In the conclusion section, it would be interesting to expand more on the practical implications of the results (i.e. how these results concerning high probability of elaboration might be concretely used to frame campaigns? Etc…)
Minor issues:
- Please double check the language (i.e. line 6) throughout the entire manuscript
Author Response
Dear R1,
It is with excitement that we resubmit to you a revised version of our manuscript. We appreciate the time and detail provided by you and have incorporated the suggested changes into the manuscript to the best of our ability. The manuscript has certainly benefited from these insightful revision suggestions. We look forward to working with you to move this manuscript closer to publication in Behavioral Sciences. You will find all revisions attached. |

Reviewer 2 Report
The manuscript reports findings of an interesting study examining the relationships of the need for cognition, two personality traits (openness to experience and agreeability) with belief in conspiracy and scientific theories. This is a timely study on understanding the psychosocial correlates of endorsement of conspiracy theories, as well as the underlying mechanisms of development and maintenance. The findings have potential real-world implications, especially in the current time when conspiracies are common in daily life. The strengths of the study are the use of validated scales to measure key variables and steps to pilot test the stimuli before the main study. I have a few comments for the authors’ consideration, as follows:
Introduction:
- A definition of conspiracy theories (and probably a few examples of COVID-related conspiracy beliefs) would be helpful to be included in the 1st paragraph, to ensure all readers have the same understanding of the topic.
- In section 2, the authors discussed beliefs and attitudes, extending into the theoretical formulation of endorsement of conspiracy theories. My understanding is that “attitude” has an affective component (e.g. liking / disliking an idea/ a belief). However, the focus of this study is mainly on beliefs in conspiracy and scientific theories (ref section 3.2.2), without touching on the liking/ affective component. The terms “beliefs” and “attitudes”, which should be well differentiated, were used interchangeably in this manuscript. I recommend making it clear and explicit whether “beliefs” or “attitudes” is the focus, and using that term consistently throughout the manuscript. This would make the Introduction, Method and Discussion will be more aligned.
- I am not very sure about H1, which seems to me that the level of belief in a particular posting is quite specific to the content of the posting itself, rather than a general phenomenon. I wonder about the relevance of H1, which could be left out, without discrediting the value of the work. Or the authors should either rephrase that or downplay it without making it a formal hypothesis.
- Section 2.3.4 (especially lines 160 – 175 p. 4) provided a summary of the inconsistent evidence for the association between personality traits and beliefs in conspiracy theories. I would like to know more about the theoretical arguments for the positive and negative associations between them, which is a theoretical framework of this study. That is, are there any speculations/ theoretical backups regarding the positive and also the negative association between agreeableness (and openness to experience) and belief in conspiracy theories? Definitions of agreeableness and openness to experience would be highly informative to help readers understand these relationships better, and should be offered here. In addition, a reference is needed for the statement “the association between agreeableness and conspiracy beliefs was higher among samples which included a larger proportion of older participants” (p. 4, lines 174-175).
Methods:
- The term “credibility” is used in several places in the manuscript. It is a bit confusing, as in this context, the term could refer to the source credibility of the postings in the preliminary study (section 3.2.1), and one of the items to measure belief in conspiracy theories & scientific theories (line 235, p. 6, section 3.2.2). The term is somehow used interchangeably with beliefs (ref title of Table 2 vs Table 3, and also Figure 1). Please be consistent about the use of the term, or clarify the use of the term where appropriate.
- I appreciated the a-prior power analysis conducted very much. However, the current sample (N = 175) was much larger than suggested by the power analysis (N = 89). Is there any reason to recruit a sample much larger than suggested by the power analysis?
- The step(s)/ procedure(s) of quality control used (line 309, p. 7) should be mentioned.
- In line 324, the “employed” is redundant.
Results:
- I suggest a re-organization of this section (especially sections 4.2 and 4.3) so that the findings would be presented in line with the sequence of the hypotheses and the research question, as presented in the Introduction. Please consider removing the redundant part, or making more full use of the regression analyses (for example, adjustment for demographic covariates).
- A caption that the regression analyses had controlled for demographics should be added underneath Table 2 and Table 3.
Discussion:
- The motives of beliefs in conspiracy belief (lines 409-516, p. 10) are highly relevant to the aim of this study and also the findings. Therefore, it would be good to present the theory earlier in the Introduction to prepare the readers for the research question and the linkage to the findings.
- It is interesting to know that individuals who are more open to experience are more inclined to believe in conspiracy theories. The authors may consider some of the features of conspiracies to explain this relationship (e.g. conspiracy theories are more entertaining, van Prooijen et al., 2022) and the potential linkage to personality traits.
van Prooijen, J. W., Ligthart, J., Rosema, S., & Xu, Y. (2022). The entertainment value of conspiracy theories. British Journal of Psychology, 113(1), 25-48.
- In lines 447-449 (p. 10), the authors suggest that “The sample characteristics of the sample employed in the current study are only partially correspondent with sample characteristics which are associated with a higher association between openness to experience and conspiracy beliefs.”. Some numbers or statistical tests could help substantiate this claim.
- In line 458 (p. 11), it is claimed that “Cacioppo et al. (1996) speculate about two facets of NFC referring”. What are these two facets of NFC? Please elaborate.
Limitations and outlook: Please list some examples of the “nonreactive measures and methods of indirect measurement” (lines 499 – 500, p. 11) for readers’ consideration.
Author Response
Dear R2,
It is with excitement that we resubmit to you a revised version of our manuscript. We appreciate the time and detail provided by you and have incorporated the suggested changes into the manuscript to the best of our ability. The manuscript has certainly benefited from these insightful revision suggestions. We look forward to working with you to move this manuscript closer to publication in Behavioral Sciences. You will find all revisions attached. |

Round 2
Reviewer 2 Report
As noted in my initial review, the manuscript reports on a study examining the relationships of the need for cognition, two personality traits (openness to experience and agreeableness) with beliefs in conspiracy and scientific theories. This is an interesting and timely area of study and the authors are commended for providing a thorough response to the reviewers' recommendations. I have no further recommendations.